# Fiscal Policy and Income Inequality: The Critical Role of Institutional Capacity

Manwar Hossein Malla * and Pairote Pathranarakul

Graduate School of Public Administration, National Institute of Development Administration, Bangkok 10240, Thailand; pairote@nida.ac.th
* Correspondence: manwar.mal@stu.nida.ac.th

**Abstract:** Rising income inequality has become a defining global challenge that hinders the achievement of the United Nations Sustainable Development Goals. The paper investigates the effect of fiscal policy and institutional capacity on income inequality among developed and developing countries. Applying the system Generalized Method of Moments (GMM) to control potential endogeneity for countries from 2000 to 2019, the following results have been established. The dynamic effect captured by the first lag of inequality suggests that the widening income gap is persistent in both developed and developing countries. We also find evidence that income tax is more progressive and may abate income inequality in developing countries and not in developed countries. However, taxes on goods and services were found not to impact income equalization globally. Furthermore, the findings reveal that government size, education expenditure, and health expenditure are negatively associated with income inequality in developed countries only. Public debt was observed not to influence income distribution across the world. We observed that corruption and government effectiveness do not significantly impact income distribution in developed and developing countries for institutional capacity. However, in most cases, the coefficients of the interactions between fiscal policy and institutional capacity bear the expected signs, albeit insignificant. Some policy recommendations have been offered.

**Keywords:** fiscal policy; institutions; income inequality; taxes; GMM

## 1. Introduction

There is no doubt that growing income inequality within and among countries is a defining challenge of the United Nations Sustainable Development Goals. Rising income inequality has generated much scholarly and policy attention over recent decades. Though it is a feature of low-income countries, rich countries with high economic growth have witnessed a widening income gap among citizens (Coady and Dizioli 2018; Nolan et al. 2019). For example, recent evidence from China (Cevik and Correa-Caro 2020; Ravallion and Chen 2021) and the United States of America (Tsui et al. 2016) attests to the worrying trend of widening income gap in high-income countries. Income inequalities have risen for a majority of the world's population: The World Inequality Report 2022 reveals that the wealthiest 10% of the global population currently controls 52% of global income, whereas half of the global population who are deemed to be the poorest holds 8.5% of the global income (Chancel et al. 2022, p. 10). Implicit in this widespread concern is the view that income inequality has severe implications for crime, social cohesion, political stability, poverty, and social justice, which subsequently undermine investment and life-improving public policy reforms (Jenkins 2017; Law and Soon 2020; Piketty and Saez 2003).

Conversely, scholars have insinuated that not until there is a corresponding decline in inequality between the rich and the poor there is no way poverty reduction efforts will be achieved despite high economic growth (Datt et al. 2020; Kwasi Fosu 2018; Shimeles and Nabassaga 2018). Consistent with this view, empirical evidence suggests that where

the income of 20% of the population increases, GDP growth rates plummet in the medium term. In contrast, the economy tends to achieve a high growth rate when the income share held by the bottom 20% increases (Dabla-Norris et al. 2015). Moreover, this evidence seems to feed into Kuznets's (1955) long-held view that income inequality tends to widen during the initial stages of development, but declines, as the economy grows to a certain level (Kuznet's inverted U-shape curve). This implies that income distribution matters for economic growth and poverty alleviation efforts. Ensuring even distribution of the national cake is crucial for poverty alleviation, maintenance of peace and security, and achieving good macroeconomic stability and growth (Anyanwu et al. 2016; Kunawotor et al. 2022).

Fiscal policy is widely seen as a vital policy instrument to ensure income distribution. Fiscal policy in the form of taxation and social spending influences the welfare of household members through monetary payment by way of taxes and transfers as well as through the provision of in-kind social benefits, including expenditure on free education and health care uptake (Clements et al. 2015, p. 3; Gupta 2018). However, some scholars have expressed that fiscal policy is ineffective in addressing income inequality because of its lower tax-to-GDP ratio (Kunawotor et al. 2022). The net impact is that limited expenditure will be allocated for social sectors such as education and health care services, which benefit the have-nots. Moreover, where fiscal policy takes the form of indirect tax, its consequential impact on income inequality is devastating (Apergis 2021). Accordingly, an active body of the literature (Brinca et al. 2021; Caminada et al. 2019; Cevik and Correa-Caro 2020; Dotti 2020; Salotti and Trecroci 2018) has examined the redistributive impact of fiscal policy with inconclusive results.

It should be noted that the decidedly mixed results of the nexus between fiscal policy and income inequality could stem from the influence of other factors, including institutional capacity. Yet, the existing publications have paid little or no attention to institutional quality or governance as a possible window for addressing inequalities between the rich and poor. Fiscal policy may be ineffective in achieving distributive outcomes without efficient institutional or bureaucratic capacity. Analogous to this view, Hyden (2007) has asked whether poor service delivery in social sectors in developing societies can be attributed to other factors save poor institutional or governance environment. Hu and Mendoza (2013) also argue that institutional capacity underpins whether public resources and public reforms are effectively allocated and influence redistributive outcomes. In this sense, institutional capacity is defined in this paper as the capacity of state institutions to promote economic prosperity and ensure that such property is shared among the citizens.

This paper investigates the effect of fiscal policy and institutional capacity on income inequality among developed and developing countries. The World Inequality Report 2022 reveals that the Middle East and North Africa (MENA) are the unequal regions globally, with the top 10% of the population controlling 58% of the region's income. In Europe, however, the income shares of the top 10% is 36%. Asia's 10% income share is 43%, Latin America is 55%, North America is 48%, and Sub-Saharan Africa is 56% (Chancel et al. 2022, p. 10). It should be noted that despite some regions appearing to be doing well concerning income distribution, it can be observed that income inequality is still a challenge across countries.

Nevertheless, since Europe and North American countries (developed) appear to be doing relatively better, it is imperative to examine what is driving the difference based on fiscal policy and institutional quality. This will provide policy implications for other countries, particularly developing countries. Moreso, since most economies, particularly in the developing world, are hard hit by the COVID-19 pandemic, it is expected to devastate the recovery process. The impact of COVID-19 on these countries amplifies the need for fiscal policy not only for post-COVID-19 recovery but also to achieve redistributive outcomes. Therefore, accounting for the dynamics of institutional capacity within fiscal policy and income inequality nexus is a novelty that has been glossed over in the literature. The paper makes some important contributions: first, since governments across the globe rely on fiscal policy not only to raise adequate revenue but also to spend in an economy so

as to reduce poverty and promote development, the paper leads evidence to show whether such policies have an inclusive effect. Second, the paper further provides evidence to show whether the institutions required for income redistribution support the effectiveness of fiscal policy in reducing income inequality across countries. The rest of the paper is arranged as follows: related literature is discussed, after which the methods are defined. The findings are discussed, policy implications are supplied, and concluding remarks are provided.

## 2. Literature Review

### 2.1. Fiscal Policy and Income Inequality

Fiscal policy is "the setting of the level of government spending and taxation by policymakers" (Mankiw 2021, p. 793). Theoretically, fiscal policy has implications on income distribution through the channels of taxes, public expenditure, and transfers (Salotti and Trecroci 2018). Thus, fiscal policy enhances equity plan in two ways: first, de Freitas (2012) argues that direct taxes are deemed progressive because they encourage income distribution and reduce income inequality. Thus, taxes imposed on incomes, capital, wealth, inheritance, and private properties distribute resources from the rich and super-rich to the poor and marginalized segments of the society (Odusola 2017). People in the high-income group would have to pay a more significant proportion of their income as tax. However, indirect taxes, such as taxes imposed on the consumption of goods and services are regressive since both the rich and the poor pay the same amount on goods and services as tax. Second, the impact on redistributive outcomes tends to be far-reaching if the revenues raised from taxes go to finance social spending to support the poor, vulnerable, and marginalized groups.

In recent times, some scholars have conducted empirical studies on the redistributive effect of fiscal policy on income inequality. Employing the panel data technique, Salotti and Trecroci (2018) examined the redistributive impact of fiscal policy in OECD countries. Thus, public debt, government size, public expenditure on education and social security, income tax, property tax, and taxes on goods and services were used to measure fiscal policy. They showed that a rise in public debt and expenditure encourages unequal income distribution, albeit minimally. It is further observed that public spending on education, social spending, and consumption promotes distributive effects. Moreover, income and property taxes were found to have equalizing effects. Clifton et al. (2020) investigated the impact of fiscal policy on income inequality for Latin American countries in the 2000s. The authors generally observed that fiscal policy marginally reduces income inequality. Specifically, spending on education, income taxes, and social security contributions were instrumental in decreasing income inequality. A panel data analysis (Apergis 2021) found that social transfers are more potent fiscal policy tools in abating income inequality than taxes. Odusola (2017) also established in Africa that low levels of taxes and social spending undermine the distributional impacts of fiscal policy. Kunawotor et al. (2022) have empirically demonstrated that fiscal redistribution in income taxes and transfers reduces the income gap between the rich and the poor a relatively small extent. Based on these decidedly mixed results, the following hypothesis is proposed:

**Hypothesis 1 (H1).** *There is a significant relationship between fiscal policy and income inequality in developed and developing countries.*

### 2.2. Institutional Capacity and Income Inequality

By definition, institutions are "humanly devised constraints that shape human interactions" (North 1990, p. 3). Institutions serve as a fulcrum around which everything within an economy revolves. They are described as the "rule of the game in a society" (North 1990, p. 3). Theoretically, Acemoglu et al. (2014) argue that institutions influence economic and redistributive outcomes by either being inclusive or extractive. While inclusive institutions stimulate economic activities, an institutionally extractive environ-

ment undermines economic development. The institutionalization of good governance practices stimulates economic growth and engenders redistribution and equalization of incomes through efficient allocation of resources and economic freedom (Acemoglu and Robinson 2006). An effective institutional capacity to implement and administer income redistributive policy reforms will invigorate the efficiency of fiscal policy to reduce income inequality. The rule of law and absence of corruption will engender social cohesion and ensure that social spending on education, health, and social transfers benefit the poor and marginalized groups. There is enough evidence that insufficient institutional capacity has a devastating impact on income distribution (Albertus and Menaldo 2014). As Odusola (2017, p. 170) argues, "when efficiency and quality of government spending are assured (through institutional capacity), public expenditure is a potent tool to redistribute wealth and opportunities to the lowest quintiles of the population". Law and Soon (2020) also argue that in an environment where institutional capacity is established, the opportunity for inclusive economic planning to promote income distribution is enhanced.

Sonora (2019) examined the impact of the rule of law and income inequality in Latin American countries. They found that property protection through an effective legal regime and effective control of corruption narrows the gap between the rich and poor. By employing four measures of institutional quality: government effectiveness, corruption, political rights, the rule of law, Adeleye et al. (2017) investigated the interactive effect of institutions and financial development in sub-Saharan Africa and concluded that the ability to control corruption would increase the efficiency of economic growth in the reduction of income inequality. Carmignani (2009) observed that weak institutions widen the income inequality. Huynh (2021) also examined how institutional quality and foreign direct investment influence income distribution in 36 Asian countries and concluded that institutional quality abates income inequality. The authors further observed that the quality of institutions moderates the impact of foreign direct investment on income inequality.

Similarly, Nguyen (2021) found that effective governance or institutions and expenditure on education reduce income inequality in both developed and developing countries. While the author also observed that economic growth widens the income inequality, foreign direct investment narrows it in developing countries but widens it in developed countries. This finding is astonishing because one would have thought that foreign direct investment should promote income distribution in developed countries given their good governance environment. The rationale is that under good institutional capacity in developed countries where policies and regulations are designed, formulated, and implemented, economic growth should be encouraged to achieve distributive outcomes. From the above discussion, the paper proposes the following interrelated hypotheses:

**Hypothesis 2 (H2).** *There is a significant and negative relationship between institutional quality and income inequality.*

**Hypothesis 3 (H3).** *The effect of fiscal policy on income inequality is moderated by the quality of the institutions.*

The above discussions suggest that the cumulative effect of fiscal policy and institutional capacity is inconclusive and deserves research attention. Therefore, this paper's resolve is to verify this relationship and offer empirical evidence that will guide policymakers to effectively deal with the intractable problem of income inequality, which has been described as the defining challenge of the United Nations Sustainable Development Goals.

## 3. Methodology and Data Description

### 3.1. Methodology

Panel data analysis is often beset with several constraints as omitted variable bias, measurement error, unobserved time-invariant and country-specific characteristics, autocorrelation, and endogeneity or the problem of reverse causality (Phillips and Sul 2007).

This paper draws insight from the recent literature on income inequality (Cevik and Correa-Caro 2020; Clifton et al. 2020; Kunawotor et al. 2020; Kunawotor et al. 2022; Odusola 2017; Salotti and Trecroci 2018) to deal with these challenges. Due to the persistent nature of income inequality, the empirical estimation strategy of this paper is modeled in line with this recent literature. Therefore, the empirical model for this paper suggests that income inequality is predicated on its first lag, fiscal policy measures, institutional capacity, and a set of covariates used in the policy literature as shown below:

$$\text{Inequality}_{i,t} = \text{Inequality}_{i,t-1} + \sum_{h=1}^{4} w_h W_{h,i,t-t} + \eta_i + \varepsilon_{it}$$

where Inequality is the income inequality measured by the disposable income of the Gini index of households' disposable income after taxes, liabilities, and receiving benefits. Inequality$_{it\_1}$ is the first lag of income inequality used to capture the dynamic effect. It is measured from 0 (indicating perfect equality) to 100 (shows perfect inequality). The subscript i at period t are the country and time effects. Fpolicy stands for the measures of fiscal policy variables, including public debt (pdebt), income tax (intax), government general consumption (gsize), health expenditure (helex), education expenditure (eduex), inflation (inf), and taxes on goods and services (gstax). Similarly, Institution denotes the measures of institutional quality, including government effectiveness (ge), corruption (corr), and democracy (democ). W is the vector of the control variables, $\eta_i$ is the country-specific effect, and $\varepsilon_{i,t}$ is the error term.

Since the existing literature holds that previous policies influence current social, political, and economic processes, ordinary least squares (OLS), fixed and random effects are likely to produce misleading results (Phillips and Sul 2007). In that regard, the dynamics of the robustness for income inequality were examined using a dynamic panel model with a lagged dependent variable using the system GMM. The GMM takes care of the persistent nature of the dependent variable, the omitted variable problem, measurement error, endogeneity, and country-specific heterogeneity (Arellano and Bond 1991; Arellano and Bover 1995; Blundell and Bond 1998). GMM is efficient in a condition where the number of cross-sections is greater than the number of periods (N > T). The consistency of the system GMM estimator is assessed through the Hansen test of overidentification restrictions for the overall validity of the instruments and the test for the null hypothesis where the error term is not serially correlated. In the situation of failing to reject the two hypotheses, it provides support for the model's validity (Blundell and Bond 1998; Roodman 2009). Hence, a Two-step system GMM was used to estimate the model.

In this paper, system GMM is employed for several reasons. Firstly, the cross-sections (N) are more than the time series (T). Thus, developed countries are 35, and developing countries are 33, with a sample period of 20 years. Secondly, time-invariant omitted variables can be well addressed using GMM since unobserved country-level heterogeneity can be accounted for. Third, internal instruments are used to address issues related to potential endogeneity. Since income inequality is found to be persistent both within and across countries (Adeleye et al. 2017; Kunawotor et al. 2020; Shimeles and Nabassaga 2018), the use of its first lag is made possible within GMM estimation to capture such persistency.

### 3.2. Sources of Data and Description of Variables

The paper investigates a panel of 68 developed (35) and developing (33) countries from 2000 to 2019, subject to data availability (see Appendix A). The data were sourced from reputable intergovernmental and international organizations, including income inequality data from the Standardized World Income Inequality Database (SWIID), institutional capacity data from the World Governance Indicators and Transparency International, Fiscal policy data from the International Monetary Fund (IMF) Financial Statistics, economic and demographic data from the World Development Indicators, World Bank, and democracy data from the Polity-V project. The detailed description of the variables and their sources is

shown in Table 1, and their respective statistics are shown in Table 2. The correlation matrix is operationalized in Appendix B.

**Table 1.** Variable Description.

| Data | Definition/Measurement | Source |
|---|---|---|
| Income inequality | The extent to which income is distributed among individuals or households. It is disposable income after tax. It is measured as 0 (perfect income distribution) 100 (perfect inequality) | Swiss Standardized Income Inequality Database (SWIID) |
| Government Consumption | Total government expenditure (% GDP) | International Monetary fund |
| Government Debts | Total government debt (% GDP) | International Monetary Fund |
| Direct tax (Income Tax) | Total income tax revenues (% GDP) | |
| Indirect tax (Taxes on Goods and Services) | Total revenue raised from taxes imposed on the consumption of goods and services (%GDP) | International Monetary Fund |
| Government Education Expenditure | Total expenditure on education (% GDP) | World Bank |
| Government Health Expenditure | Total expenditure on health (%GDP) | World Bank |
| Government Effectiveness | | World Governance Indicators |
| Corruption | | Transparency International |
| Democracy | Polity 2 Index. Measure from −10 (most autocratic) and +10 (most democratic) | Polity-V Project |
| Population growth rate | | World Bank |
| Foreign Direct Investment | Foreign Direct Investment (%GDP) | |
| GDP per capita | GDP per capita (constant 2010 US$) | World Bank |
| Trade openness | Total exports and imports (% GDP) | World Bank |

Source: authors' construction.

**Table 2.** Descriptive statistics.

| Variable | Obs | Mean | Std. Dev. | Min | Max |
|---|---|---|---|---|---|
| Income inequality | 1186 | 35.376 | 8.251 | 22.4 | 59.9 |
| Government size | 1253 | 16.884 | 4.607 | 4.846 | 27.935 |
| Income tax | 1149 | 24.864 | 12.659 | −1.348 | 66.715 |
| Tax on goods and services | 1147 | 33.383 | 9.327 | 6.342 | 77.688 |
| Public debt | 1234 | 50.28 | 33.845 | 0.828 | 198.438 |
| Education expenditure | 958 | 4.911 | 1.389 | 1.496 | 9.51 |
| Health expenditure | 1122 | 6.653 | 2.302 | 1.916 | 13.677 |
| Government Effectiveness | 1188 | 64.58 | 24.671 | 0 | 100 |
| Corruption | 528 | 53.741 | 19.656 | 24 | 92 |
| Democracy | 1235 | 7.241 | 4.416 | −10 | 10 |
| GDP per capita | 1251 | 23,257.744 | 17,932.356 | 1075.395 | 97,864.195 |
| Foreign Direct Invest. | 1303 | 6.914 | 25.519 | −58.323 | 451.639 |
| Inflation | 1195 | 96.637 | 24.273 | 30.76 | 261.069 |
| Trade openness | 1254 | 94.339 | 62.549 | 19.798 | 437.327 |
| Population growth | 1254 | 0.731 | 0.971 | −9.081 | 7.786 |

Our explanatory variable of interest is the fiscal policy, which is proxied by fiscal redistribution, tax measures (direct and indirect taxes), and public or government expenditure measures. First redistribution (income inequality) consists of disposable Gini coefficient after taxes. Tax indicators include income tax (direct tax measure), government consumption-based tax (indirect tax), and taxes on goods and services (indirect tax). Moreover, the indicators of public expenditure encompass expenditure on health and education and public debt situation. The spending on health and education shows investment in social sectors that may benefit the poor and marginalized groups. They (expenditure on health and education) are expressed as a percentage of GDP. This paper expects fiscal policy as direct taxes should lead to income distribution and reduce the income gap between rich and poor, while indirect taxes should widen the income gap. Two variables were used to measure Institutional capacity: government effectiveness and corruption. Government effectiveness looks at the perception of the quality of public services, the quality of civil service, and the degree of its independence from political manipulations. Corruption measures the abuse of public office for personal gains. The prior expectation of this paper is that institutional quality should have a reducing effect on income inequality. The direction of the relationship between corruption and income distribution is uncertain.

As a standard practice in the existing income inequality literature, some control variables were added to the model: population growth rate, trade openness, GDP per capita, foreign direct investment (FDI), and democracy. Population growth rate affects income distribution through age dependency ratio: thus, either youthful population between 0–15 years or old age population of 65 and above against the active working population of 16–64 years (Kunawotor et al. 2022). Therefore, we expect that age dependency should increase with income inequality. Gross domestic product (GDP) per capita measures the amount of income received by the citizens of a country. As per the Kuznets-curve hypothesis (Kuznets 1955), we expect that income inequality will increase with GDP per capita in the short term but decrease in the long run. Trade openness measures the volume of exports and imports. Trade openness is related to the dynamics of trade liberalization. Thus, trade liberation can open economic opportunities for low-skill and low-income people, likely reducing income inequality. However, trade liberalization may also open the domestic economy to external shocks resulting in domestic volatilities. Therefore, the relationship between trade openness and income distribution is uncertain (Dabla-Norris et al. 2015). Moreover, the net inflow of foreign direct investment is used to measure FDI. However, we assume FDI to exert a negative impact on income inequality. Democracy is controlled because it is only within a democracy that citizens are offered the opportunity to exercise a relatively high degree of control over leaders to ensure their needs and preferences are redressed. Therefore, democracy is responsive to the poor and vulnerable groups, which may contribute to the abatement of income inequality (Reuveny and Li 2003).

## 4. Empirical Results and Discussions

This paper examines the cumulative effect of fiscal policy and institutional capacity on income inequality with evidence from both developed and developing countries. Therefore, some steps were followed: first, the model involving fiscal policy measures was examined; and second, the following model investigated the interactive effect of fiscal policy and institutional capacity on inequality. It should be underscored that negative signs on the coefficients indicate the potential of the predictor variables to reduce income inequality. In contrast, positive coefficients show the potential of the variables to widen the income gap between the rich and the poor.

### 4.1. Empirical Results of the Effect of Fiscal Policy on Income Inequality

The empirical findings of the redistributive impacts of fiscal policy encompass the tax measures and expenditure dimensions. Tables 3 and 4 are the baseline models and show the empirical results of the effects of fiscal policy on income inequality for developed and developing countries, respectively. While tax variants (income tax and general taxes

on goods and services) of fiscal policy are presented in Model (1)–Model (2) that of the expenditure dimensions (government size, expenditure on education and health, and public debt) are shown in Model (3)–Model (6) respectively. The results (Model (1)– Model (6)) reveal that the coefficients of the dynamic effects (first lag of inequality) are more than 0.800 and statistically significant at 1% (Tables 3 and 4), which imply that income inequality is persistent in both developed and developing countries. Adeleye et al. (2017) have argued that the lagged value greater than 0.800 is required as the rule of thumb for establishing such persistence. It also means that the past income inequality level is a stronger determinant of its current level. The dynamic effect further indicates that income inequality tends to be path-dependent, as the current income inequality of a country strongly predicts her level of income inequality in the ensuing year. This is consistent with the findings in the existing literature (Adeleye et al. 2017; Anyanwu et al. 2016; Kunawotor et al. 2022).

**Table 3.** Effect of fiscal policy on income inequality (developed countries).

| Variables | Model 1 | Model 2 | Model 3 | Model 4 | Model 5 | Model 6 |
|---|---|---|---|---|---|---|
| inequality-1 | 1.024 *** (0.0706) | 1.033 *** (0.0830) | 0.994 *** (0.0781) | 1.014 *** (0.0536) | 0.859 *** (0.165) | 0.958 *** (0.0632) |
| intax | −0.00185 (0.00544) | | | | | |
| gstax | | 0.00424 (0.00893) | | | | |
| gsize | | | 0.0348 * (0.0192) | | | |
| eduex | | | | 0.187 * (0.105) | | |
| helex | | | | | 0.0877 * (0.0458) | |
| pdebt | | | | | | $-1.67 \times 10^{-5}$ (0.00115) |
| gdpc | $-6.23 \times 10^{-6}$ ** $(2.92 \times 10^{-6})$ | $-7.03 \times 10^{-6}$ ** $(2.71 \times 10^{-6})$ | $-1.61 \times 10^{-6}$ $(3.90 \times 10^{-6})$ | $-8.20 \times 10^{-6}$ $(9.58 \times 10^{-6})$ | $-2.56 \times 10^{-5}$ $(2.36 \times 10^{-5})$ | $-1.02 \times 10^{-5}$ *** $(3.18 \times 10^{-6})$ |
| gdppc square | | | | | $1.44 \times 10^{-10}$ $(1.81 \times 10^{-10})$ | |
| fdi | −0.000297 (0.000865) | −0.000380 (0.000824) | $-8.44 \times 10^{-5}$ (0.000682) | −0.000178 (0.000328) | 0.000292 (0.000415) | −0.000229 (0.000741) |
| to | −0.000502 (0.00130) | −0.000426 (0.00138) | 0.000488 (0.00156) | 0.000309 (0.00147) | 0.00184 (0.00212) | $7.49 \times 10^{-5}$ (0.00153) |
| popg | 0.0173 (0.0268) | 0.0147 (0.0263) | 0.00980 (0.0204) | 0.0394 * (0.0226) | 0.0514 ** (0.0248) | 0.0256 (0.0324) |
| inf | 0.000306 (0.00177) | 0.000138 (0.00170) | −0.00189 (0.00179) | −0.00264 (0.00322) | −0.00290 (0.00293) | 0.000961 (0.00265) |
| democ | 0.0560 (0.0683) | 0.0415 (0.0672) | 0.0497 (0.0713) | 0.0648 (0.134) | 0.0458 (0.0859) | 0.00318 (0.0489) |
| Observations | 537 | 537 | 542 | 446 | 507 | 544 |
| Number of groups | 34 | 34 | 34 | 34 | 34 | 34 |
| Number of instru. | 26 | 26 | 25 | 24 | 25 | 25 |
| AR(1) | 0.008 | 0.010 | 0.009 | 0.019 | 0.066 | 0.014 |
| AR(2) | 0.253 | 0.246 | 0.258 | 0.299 | 0.362 | 0.286 |
| Hansen J. | 0.581 | 0.658 | 0.656 | 0.606 | 0.700 | 0.531 |

Standard errors in parentheses. *** $p < 0.01$, ** $p < 0.05$, * $p < 0.1$.

The results (Model 1) reveal that direct tax (income tax) is negative but statistically insignificant for developed countries (Table 3), suggesting that income tax has the potential to reduce inequality. However, concerning developing countries (Table 4), income tax is negative and statistically significant at a 5% level. Thus, a unit increase in the level of direct tax, mainly income tax, is likely to reduce income inequality by 0.0118. This implies that direct taxes in the form of income tax are powerful fiscal policy tools to reduce income inequality in developing countries. This is particularly so because income tax is classified as a progressive tax where the tax rate increases as the income increases. With income tax, the wealthy or high-income people bear the tax burden by paying a more significant

proportion of their income as tax. This result is consistent with the work of Clifton et al. (2020); Kunawotor et al. (2022); and Salotti and Trecroci (2018).

**Table 4.** Effect of fiscal policy on income inequality (developing countries).

| Variables | Model 1 | Model 2 | Model 3 | Model 4 | Model 5 | Model 6 |
|---|---|---|---|---|---|---|
| inequality-1 | 0.912 *** | 0.910 *** | 0.904 *** | 0.988 *** | 0.939 *** | 0.897 *** |
| | (0.0384) | (0.0479) | (0.0608) | (0.0526) | (0.0347) | (0.0392) |
| intax | $-0.0118$ ** | | | | | |
| | (0.00525) | | | | | |
| gstax | | 0.00166 | | | | |
| | | (0.00705) | | | | |
| gsize | | | $-0.00250$ | | | |
| | | | (0.0152) | | | |
| eduex | | | | 0.00528 | | |
| | | | | (0.0320) | | |
| helex | | | | | $-0.0195$ | |
| | | | | | (0.0477) | |
| pdebt | | | | | | 0.00211 |
| | | | | | | (0.00221) |
| gdppc | $-3.94 \times 10^{-5}$ *** | $-3.88 \times 10^{-5}$ * | $-4.45 \times 10^{-5}$ * | $-1.48 \times 10^{-5}$ | $-3.61 \times 10^{-5}$ | $-3.70 \times 10^{-5}$ * |
| | $(1.39 \times 10^{-5})$ | $(2.21 \times 10^{-5})$ | $(2.46 \times 10^{-5})$ | $(1.34 \times 10^{-5})$ | $(2.15 \times 10^{-5})$ | $(2.01 \times 10^{-5})$ |
| fdi | 0.00596 | 0.00207 | 0.00117 | 0.00164 | 0.00208 | 0.00280 |
| | (0.00558) | (0.00421) | (0.00348) | (0.00301) | (0.00335) | (0.00323) |
| to | $-0.00133$ | $-0.00176$ | $-0.00261$ | $-0.00214$ | $-0.00153$ | $-0.00218$ |
| | (0.00154) | (0.00225) | (0.00229) | (0.00229) | (0.00176) | (0.00169) |
| popg | 0.0502 | 0.0734 | 0.139 | 0.169 | 0.114 | 0.112 |
| | (0.0973) | (0.102) | (0.114) | (0.104) | (0.0833) | (0.0784) |
| inf | 0.000449 | $-0.000306$ | $-0.000378$ | $-0.000897$ | $-0.00127$ | $-0.00102$ |
| | (0.00105) | (0.00109) | (0.000709) | (0.000794) | (0.00117) | (0.00127) |
| democ | $-0.000669$ | 0.000189 | 0.00166 | 0.000796 | 0.00417 | 0.00362 |
| | (0.00441) | (0.00474) | (0.00545) | (0.00583) | (0.00528) | (0.00523) |
| Observations | 397 | 395 | 402 | 308 | 430 | 454 |
| Number of groups | 30 | 30 | 30 | 29 | 30 | 30 |
| Number of instru. | 25 | 26 | 25 | 24 | 23 | 25 |
| AR (1) OIR | 0.017 | 0.033 | 0.026 | 0.065 | 0.028 | 0.017 |
| AR(2) OIR | 0.070 | 0.080 | 0.082 | 0.112 | 0.238 | 0.070 |
| Hansen | 0.545 | 0.329 | 0.233 | 0.411 | 0.618 | 0.545 |

Standard errors in parentheses. *** $p < 0.01$, ** $p < 0.05$, * $p < 0.1$.

However, though indirect tax (taxes on goods and services) is statistically insignificant for both developed and developing countries, the coefficients are positive, suggesting that indirect taxes such as taxes on the consumption of goods and services have the potential to widen the gap between the rich and the poor in both developed and developing countries. More interestingly, the coefficients of the expenditure dimensions of fiscal policy (Table 3), including the size of the government, public debt, public expenditure on education and health, are positive and statistically significant at 10% levels (Model 3, Model 4 and Model 5). This implies that government size, public investment in education, and health tend to increase income inequality in developed countries (Table 3). Thus, a unit increase in the size of the government will correspondingly increase income inequality by 0.0348. Moreover, the income gap between the rich and the poor in developed countries is also likely to widen by 0.187 and 0.0877 when government expenditure on education and health increases by a unit each, respectively. This result is astonishing in the sense that it is not clear the condition under which government expenditure on social sectors tends to widen the income gap between the rich and the poor in developed countries. The findings contradict the existing literature (Apergis 2021; Gupta 2018; Odusola 2017). However, the findings concerning developing countries show that the expenditure dimensions of fiscal policy have no significant impact on income inequality.

It can further be observed that GDP per capita reduces income inequality in developed and developing countries. Model (4) and Model (5) in Table 3 signal that population growth increases income inequality in developed countries but does not seem to impact developing countries. However, foreign direct investment, inflation, trade openness, and democracy

do not affect income inequality in developed and developing countries. Given the choice of one lag length, the specification of the second-order Arellano and Bond autocorrelation test (AR(2)) results reveal that the system GMM model does not suffer from second-order serial correlation, and the Hansen J-test of overidentification restrictions shows that the instruments used are not over-identified. Therefore, the system GMM model specification fits and can be relied upon for subsequent discussion and policy extrapolation.

*4.2. The Moderating Role of Institutional Capacity in Fiscal Policy-Income Inequality Nexus*

Having accounted for the impact of fiscal policy on income inequality in our Baseline Models (Tables 3 and 4), this section examines whether institutional quality moderates the effect of fiscal policy on income inequality. While Table 5 shows the results for developed countries, Table 6 presents developing countries' results. From Tables 5 and 6, Model (1)–Model (6) show the measures of institutional capacity and their interactions with the variants of fiscal policy. Thus, the models show the role of institutions in fiscal policy and the income inequality nexus captured by interacting with the measures of institutional capacity and fiscal policy. We expect that the coefficients of the interactions between corruption and fiscal policy measures should be positive to signal the domineering impact of corruption in increasing income inequality. In contrast, the coefficients of the interactions between government effectiveness and fiscal policy measures are expected to be negative to reduce income inequality.

As shown (Tables 5 and 6), the findings generally reveal no significant effect of institutions on income inequality in both developed and developing countries. Though the interaction terms are also not statistically significant, they bear expected signs in most cases, which suggest that institutions have an equalizing effect on income inequality if they are significant. Nevertheless, some salient points can be deduced from these results. The insignificance of institutional capacity in developed countries can be explained through institutional inertia (Madni 2019), which shows that the slow changes in the institutional environment are somewhat responsible for unequal income distribution. The institutional inertia argues that at a particular stage of the institutional development, new institutions need to be created to deal with unknown causes of income inequality since the exiting institutions may act slow or may not be enough to deal with emerging causes of the unequal income distribution (Easton 1965; Josifidis et al. 2017). Therefore, institutional inertia may explain why developed countries experience rising income inequality despite high economic growth and matured institutional environment. However, the lack of statistical strength on the part of institutional capacity concerning developing countries may be due to the relatively weak nature of institutions and poor governance environment.

Despite the insignificance of corruption and its interactions with fiscal policy, it does not suggest that it is irrelevant. Corruption may undermine the opportunity for income distribution in two ways. First, since people with high income and well-connected to political elites are the beneficiaries of corruption, the ability of the government and its allied institutions to ensure even distribution of economic resources may be neutralized (Furceri and Ostry 2019). Second, concerning taxation, because corruption benefits the rich and the most connected people, induces a biased tax regime. Subsequently, tax evasion through corruption undermines the government's capacity to raise enough revenue talks less of distributing it. Third, corruption denies the poor and marginalized groups access to critical public services such as education and health. Also, the government effectiveness, which implies invigorating the quality of public services, the effectiveness of social policies and programs is crucial in reducing income inequality in developed and developing countries.

**Table 5.** Moderating role of fiscal policy and institutional capacity on income inequality (developed countries).

| Variables | Model 1 | Model 2 | Model 3 | Model 4 | Model 5 | Model 6 |
|---|---|---|---|---|---|---|
| inequality-1 | 0.482 (0.325) | 0.884 *** (0.168) | 0.731 *** (0.191) | 0.738 *** (0.191) | 0.838 *** (0.229) | 0.884 *** (0.152) |
| pdebt × corr | | | | | | 0.000114 (0.000571) |
| pdebt × ge | | | | | | −0.000897 (0.000782) |
| helex × corr | | | | | 0.0129 (0.0117) | |
| helex × ge | | | | | −7.03 × 10$^{-6}$ (0.0109) | |
| eduex × corr | | | | 0.00926 (0.0496) | | |
| eduex × ge | | | | −0.00973 (0.0347) | | |
| gsize × corr | | | 0.000623 (0.0156) | | | |
| gsize × ge | | | −0.00458 (0.00967) | | | |
| gstax × corr | | 0.00135 (0.00408) | | | | |
| gstax × ge | | −0.000183 (0.00297) | | | | |
| intax × corr | 0.00782 (0.00690) | | | | | |
| intax × ge | −0.000281 (0.00216) | | | | | |
| pdebt | | | | | | −0.0738 (0.0546) |
| gsize | | | 0.259 (1.228) | | | |
| helex | | | | | 0.896 (1.158) | |
| eduex | | | | −0.407 (5.584) | | |
| gstax | | 0.0604 (0.376) | | | | |
| intax | −0.578 (0.605) | | | | | |
| ge | 0.0605 (0.0684) | −0.0518 (0.100) | 0.0568 (0.173) | −0.0966 (0.259) | −0.0332 (0.116) | −0.121 (0.107) |
| corr | −0.0822 (0.116) | 0.0952 (0.150) | 0.120 (0.352) | 0.103 (0.326) | 0.155 (0.113) | 0.0310 (0.0598) |
| gdppc | −2.36 × 10$^{-5}$ (2.25 × 10$^{-5}$) | −1.98 × 10$^{-5}$ (1.29 × 10$^{-5}$) | −5.91 × 10$^{-5}$ (4.33 × 10$^{-5}$) | −3.73 × 10$^{-5}$ (9.48 × 10$^{-5}$) | −2.79 × 10$^{-5}$ (3.22 × 10$^{-5}$) | −1.36 × 10$^{-5}$ (1.98 × 10$^{-5}$) |
| fdi | 0.00146 (0.00267) | 0.00221 (0.00229) | 0.00273 (0.00346) | −0.00216 (0.00192) | 0.00189 (0.00185) | 0.000495 (0.00123) |
| to | 0.00660 (0.00719) | −0.001000 (0.00299) | −0.00482 (0.00456) | −0.00197 (0.00693) | −0.00170 (0.00505) | −0.00154 (0.00315) |
| popg | −0.436 (0.330) | −0.124 (0.143) | −0.274 (0.441) | −0.281 (0.302) | −0.195 (0.161) | −0.0645 (0.114) |
| inf | −0.00388 (0.0187) | −0.00725 (0.0142) | 0.0186 (0.0243) | 0.00174 (0.0504) | 0.0131 (0.0199) | −0.0203 (0.0184) |
| democ | −0.245 (0.322) | −0.00588 (0.918) | 0.853 (1.400) | −0.220 (0.712) | −0.0327 (0.863) | −0.312 (0.513) |
| Observations | 170 | 170 | 170 | 113 | 134 | 176 |
| Number groups | 34 | 34 | 34 | 32 | 34 | 34 |
| Number of instruments | 25 | 25 | 25 | 24 | 25 | 25 |
| AR(1) | 0.048 | 0.081 | 0.072 | 0.008 | 0.053 | 0.043 |
| AR(2) | 0.245 | 0.473 | 0.464 | 0.231 | 0.349 | 0.679 |
| Hansen | 0.420 | 0.442 | 0.368 | 0.157 | 0.329 | 0.298 |

Standard errors in parentheses. *** $p < 0.01$.

**Table 6.** Moderating role of fiscal policy and institutional capacity on income inequality (developing countries).

| Variables | Model 1 | Model 2 | Model 3 | Model 4 | Model 5 | Model 6 |
|---|---|---|---|---|---|---|
| inequality-1 | 0.705 *** | 0.873 *** | 0.673 *** | 0.917 *** | 0.783 *** | 0.715 *** |
| | (0.152) | (0.107) | (0.171) | (0.146) | (0.120) | (0.101) |
| pdebt × corr | | | | | | 0.000251 |
| | | | | | | (0.000240) |
| pdebt × ge | | | | | | $-9.51 \times 10^{-5}$ |
| | | | | | | (0.000222) |
| helex × corr | | | | | 0.0192 | |
| | | | | | (0.0246) | |
| helex × ge | | | | | 0.00294 | |
| | | | | | (0.00674) | |
| eduex × corr | | | | −0.00343 | | |
| | | | | (0.0202) | | |
| eduex × ge | | | | −0.00314 | | |
| | | | | (0.0125) | | |
| gsize × corr | | | 0.00331 | | | |
| | | | (0.00463) | | | |
| gsize × ge | | | −0.0120 | | | |
| | | | (0.0102) | | | |
| gstax × corr | | 0.000704 | | | | |
| | | (0.00110) | | | | |
| gstax × ge | | $-8.54 \times 10^{-5}$ | | | | |
| | | (0.000855) | | | | |
| intax × corr | 0.00232 | | | | | |
| | (0.00151) | | | | | |
| intax×ge | −0.000119 | | | | | |
| | (0.00144) | | | | | |
| pdebt | | | | | | −0.00535 |
| | | | | | | (0.00878) |
| helex | | | | | −0.773 | |
| | | | | | (1.009) | |
| eduex | | | | 0.373 | | |
| | | | | (0.458) | | |
| gsize | | | 0.599 | | | |
| | | | (0.488) | | | |
| gstax | | −0.0345 | | | | |
| | | (0.0482) | | | | |
| intax | −0.120 | | | | | |
| | (0.0929) | | | | | |
| ge | −0.0122 | −0.00105 | 0.196 | 0.0221 | −0.0253 | 0.0116 |
| | (0.0360) | (0.0308) | (0.162) | (0.0532) | (0.0430) | (0.0255) |
| corr | −0.0743 | −0.0361 | −0.0895 | 0.0304 | −0.126 | −0.0235 |
| | (0.0695) | (0.0553) | (0.102) | (0.0951) | (0.154) | (0.0302) |
| gdppc | $-4.57 \times 10^{-5}$ | $-3.39 \times 10^{-5}$ | $-6.40 \times 10^{-6}$ | $-4.59 \times 10^{-5}$ | $-7.13 \times 10^{-5}$ | $-3.89 \times 10^{-5}$ |
| | $(4.61 \times 10^{-5})$ | $(3.80 \times 10^{-5})$ | $(7.08 \times 10^{-5})$ | $(8.87 \times 10^{-5})$ | $(5.47 \times 10^{-5})$ | $(3.43 \times 10^{-5})$ |
| fdi | −0.00351 | −0.00175 | 0.00275 | 0.00126 | −0.00967 | −0.000292 |
| | (0.00631) | (0.00505) | (0.00739) | (0.00546) | (0.0114) | (0.00554) |
| to | −0.00396 | −0.000241 | 0.00446 | −0.00148 | 0.00406 | 0.00118 |
| | (0.00775) | (0.00460) | (0.0101) | (0.00725) | (0.00644) | (0.00369) |
| popg | −0.237 | 0.174 | 0.845 | 0.282 | 0.0831 | 0.265 |
| | (0.281) | (0.312) | (0.795) | (0.430) | (0.413) | (0.184) |
| inf | 0.00189 | 0.00184 | −0.00222 | 0.00238 | −0.000850 | −0.000248 |
| | (0.00317) | (0.00195) | (0.00317) | (0.00310) | (0.00285) | (0.00209) |
| democ | 0.0173 | −0.0166 | −0.000352 | −0.0285 | −0.0174 | −0.00218 |
| | (0.0207) | (0.0157) | (0.0315) | (0.0684) | (0.0206) | (0.0150) |
| Observations | 119 | 119 | 119 | 83 | 106 | 131 |
| Number of groups | 26 | 26 | 26 | 23 | 28 | 27 |
| Number of instruments | 25 | 25 | 25 | 26 | 23 | 25 |
| AR (1) | 0.335 | 0.123 | 0.258 | 0.231 | 0.112 | 0.134 |
| AR(2) | 0.128 | 0.308 | 0.408 | 0.136 | 0.996 | 0.214 |
| Hansen | 0.557 | 0.518 | 0.528 | 0.396 | 0.598 | 0.606 |

Standard errors in parentheses. *** $p < 0.01$.



## 5. Concluding Remarks and Policy Implications

In recent years, income inequality has been rising in developing countries and developed countries despite the ever-increasing level of economic growth across the globe. Extant literature has examined the drivers of income inequality, including fiscal policy. However, the empirical evidence on fiscal policy-income inequality nexus has not only produced decidedly mixed results so far; however, little attention has been paid to the role of institutional quality. Therefore, the paper examines the cumulative effect of fiscal policy and institutional capacity on income inequality in developed and developing countries. The fiscal policy measure employed includes direct tax (income tax), indirect tax (taxes on goods and services), government size, public expenditure on education and health, and public debt. Moreover, government effectiveness and corruption were used to proxy for institutional capacity. The following results have been established by applying a more robust econometric technique (system GMM) to control for potential endogeneity. The dynamic effect captured by the first lag of inequality suggests that the widening income gap is persistent in both developed and developing countries. We also find evidence that income tax is more progressive and may abate income inequality in developing countries and not in developed countries. However, taxes on goods and services were found not to impact income equalization across the globe.

Furthermore, the findings reveal that government size, education expenditure, and health expenditure are negatively associated with income inequality in developed countries only. Public debt was observed not to influence income distribution across the world. We observed that corruption and government effectiveness do not significantly impact income distribution in developed and developing countries for institutional capacity. However, in most cases, the coefficients of the interactions between fiscal policy and institutional capacity bear the expected signs, albeit insignificant. This implies that institutions play a minor role in influencing the effect of fiscal policy on income inequality.

Therefore, the paper offers the following policy recommendations based on the findings above. The paper recommends that developed and developing countries vigorously pursue tax reforms to broaden the tax net to make the structure of tax more progressive as a potent avenue to reduce income inequality. This implies that efforts need to be made to roll out more direct taxes, such as property tax. This will ensure that the rich or high-income earners pay more of their income as tax. Moreover, governments need to strengthen tax administration capacity to ensure that direct tax instruments are used more than indirect tax instruments. We emphasize that revenues accrued from taxes should be invested in social sectors such as free education and health care services to benefit the poor and marginalized segments of the population. Thus, free health insurance programs should be implemented to ensure unfettered access to health care services by the poor. Additionally, conscious efforts should be put in place to use fiscal policy as an instrument to promote growth that creates jobs ensure skills development and general improvement in human development. Finally, since developed countries may likely have reached the stage of institutional inertia, which makes it impossible to address emerging income inequality, it is crucial to create new effective institutions to achieve income equalization effects. Similarly, developing countries need to embark upon institutional renewal to ensure practical institutionalization of administrative efficiency and good governance practices. This will provide the suitable institutional capacity to address corruption and facilitate redistributive outcomes.

The paper is not without limitations. It should be noted that only government effectiveness and control of corruption are used as a measure of institutional capacity. Further studies should employ other indicators such as the rule of law, regulatory quality, and voice and accountability to very this relationship. Moreover, the study adopted the methodology that treated all countries as a single unit, likely to be glossed over individual country heterogeneities. Therefore, further studies should provide evidence on this relationship based on individual country realities.

**Author Contributions:** Conceptualization, M.H.M. and P.P.; methodology, M.H.M., and P.P.; software, M.H.M.; validation, P.P.; formal analysis, M.H.M.; investigation, M.H.M.; resources, M.H.M.; data curation, M.H.M.; writing—original draft preparation, M.H.M.; writing—review and editing, M.H.M. and P.P.; visualization, M.H.M.; supervision, P.P. All authors have read and agreed to the published version of the manuscript.

**Funding:** This research received no external funding.

**Data Availability Statement:** The data used in this study are publicly available in respective organization's websites mentioned in Table 1. Variable Description.

**Conflicts of Interest:** The authors declare no conflict of interest.

**Appendix A**

*Appendix A.1. Developed Countries*

Australia, Austria, Belgium, Chile, Croatia, Cyprus, Czech Republic, Denmark, Estonia, Finland, France, Germany, Greece, Hungary, Iceland, Ireland, Israel, Italy, Japan, Latvia, Lithuania, Luxemburg, Malta, Netherlands, New Zealand, Norway, Poland, Portugal, Singapore, Slovak Republic, Slovenia, Spain, Sweden, Switzerland, and United Kingdom.

*Appendix A.2. Developing Countries*

Burkina Faso, Nepal, Bangladesh, Bhutan, Bolivia, Cote D'Ivoire, El Salvador, India, Indonesia, Kyrgyz Republic, Moldova, Mongolia, Ukraine, Philippines, Tunisia, Brazil, Belarus, Argentina, Costa Rica, Columbia, China, Jamaica, Georgia, Dominican Republic, Peru, Malaysia, Mexica, Mauritius, South Africa, Romania, Russia, Sri Lanka, and Thailand.

**Appendix B**

*Correlation Matrix*

| Correlations Variables | (1) | (2) | (3) | (4) | (5) | (6) | (7) | (8) | (9) | (10) | (11) | (12) | (13) | (14) | (15) |
|---|---|---|---|---|---|---|---|---|---|---|---|---|---|---|---|
| (1) gini | 1.000 | | | | | | | | | | | | | | |
| (2) gsize | −0.521 | 1.000 | | | | | | | | | | | | | |
| (3) intax | 0.141 | −0.034 | 1.000 | | | | | | | | | | | | |
| (4) gstax | 0.146 | −0.088 | −0.284 | 1.000 | | | | | | | | | | | |
| (5) pdebt | 0.002 | 0.143 | 0.114 | −0.156 | 1.000 | | | | | | | | | | |
| (6) eduex | −0.332 | 0.641 | 0.046 | −0.067 | −0.006 | 1.000 | | | | | | | | | |
| (7) helex | −0.444 | 0.654 | 0.029 | −0.005 | 0.135 | 0.530 | 1.000 | | | | | | | | |
| (8) ge | −0.326 | 0.422 | 0.287 | −0.093 | 0.150 | 0.228 | 0.401 | 1.000 | | | | | | | |
| (9) cpi | −0.459 | 0.542 | 0.274 | −0.159 | 0.108 | 0.419 | 0.596 | 0.679 | 1.000 | | | | | | |
| (10) democ | −0.152 | 0.282 | 0.145 | 0.082 | 0.071 | 0.275 | 0.474 | 0.397 | 0.376 | 1.000 | | | | | |
| (11) gdppc | −0.567 | 0.378 | 0.240 | −0.257 | 0.044 | 0.321 | 0.496 | 0.628 | 0.810 | 0.280 | 1.000 | | | | |
| (12) fdi | −0.124 | 0.038 | 0.028 | 0.020 | 0.094 | 0.094 | 0.059 | 0.124 | 0.070 | 0.054 | 0.120 | 1.000 | | | |
| (13) inf | −0.092 | 0.024 | 0.067 | −0.040 | 0.056 | 0.148 | 0.178 | 0.011 | −0.481 | 0.012 | 0.108 | −0.013 | 1.000 | | |
| (14) to | −0.294 | 0.024 | −0.007 | −0.105 | 0.008 | 0.069 | −0.062 | 0.230 | 0.327 | −0.090 | 0.495 | 0.273 | 0.009 | 1.000 | |
| (15) popg | 0.375 | −0.241 | 0.323 | −0.218 | −0.012 | −0.016 | −0.236 | −0.096 | −0.041 | −0.171 | −0.003 | 0.013 | 0.039 | 0.042 | 1.000 |

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
