# Peer review of "Fiscal Policy and Income Inequality: The Critical Role of Institutional Capacity"

_economies, doi:10.3390/economies10050115_

Round 1
Reviewer 1 Report
I find the topic of the reviewed paper, entitled Fiscal policy and income inequality: The critical role of institutional capacity, crucial and compelling. The author(s) did a careful literature review using the latest research. The research is conducted correctly. The research methods used should be considered correct. The length of the research period is sufficient. However, the quality of this paper could be improved.
Below there is a list of my critical remarks on the reviewed paper:
- The aim of the study, i.e., that “paper investigates the effect of fiscal policy and institutional capacity on income inequality among developed and developing countries”, should be presented literally in the abstract.
- Please present your paper contribution a bit broader.
- Please formulate research hypothesis/es.
- I recommend presenting briefly in the conclusion section the study's limitations.
Overall assessment
The quality of this study is relatively high. I find the reviewed paper valuable and interesting, and its topic is up-to-date. I recommend making changes (considering the remarks mentioned above) that could improve the quality of the reviewed paper.
Author Response
Dear Reviewer,
We are indeed grateful to the Editor and the anonymous reviewers for their insightful comments that have enriched our manuscript.
In the main text, we have duly revised the paper following the reviewers’ comments. The revised portions are in track changes marked RED for your convenience. The details of how we have addressed the comments are as follows:
Comments |
Responses |
Reviewer #1 |
|
I find the topic of the reviewed paper, entitled Fiscal policy and income inequality: The critical role of institutional capacity, crucial and compelling. The author(s) did a careful literature review using the latest research. The research is conducted correctly. The research methods used should be considered correct. The length of the research period is sufficient. However, the quality of this paper could be improved. |
We thank the reviewer for this interesting commendation. |
Below there is a list of my critical remarks on the reviewed paper: |
|
The aim of the study, i.e., that “paper investigates the effect of fiscal policy and institutional capacity on income inequality among developed and developing countries”, should be presented literally in the abstract. |
Thank you for this comment. We have addressed this comment in the abstract. Please, see page 1 of the main text. |
Please present your paper contribution a bit broader.
|
We have addressed this comment as follows:
The paper makes several significant contributions. First, since governments across the globe rely on fiscal policy not only to raise adequate revenue but also to spend to reduce poverty and promote development, the paper leads evidence to show whether such policies have an inclusive effect. Second, the paper further provides evidence to ascertain whether the institutions required for income redistribution support the effectiveness of fiscal policy in reducing income inequality across countries. Please, see pages 2-3 of the main text. |
Please formulate research hypothesis/es.
|
Thank you for this comment. We have formulated the hypotheses and added them to the literature review section. Please, see pages 3 and 4 for details. |
I recommend presenting briefly in the conclusion section the study's limitations.
|
We are grateful for this comment. The limitations have been included in the conclusion section as requested. Please, see page 15. |
For your convenience, the article related information is provided below:
Journal: Economies
Manuscript ID: economies-1667406
Title: Fiscal policy and income inequality: The critical role of
institutional capacity
Authors: Manwar Hossein Malla *, Pairote Pathranarakul
Regards,
Manwar Hossein Malla
Deputy Secretary, Finance Division
Ministry of Finance
Bangladesh Secretariat
Dhaka 1000.
Cell Phone: +8801713123278
E-Mail: manwarhm@yahoo.com
Please see the attachment (PDF File)

Reviewer 2 Report
The research is of interest because it examines the moderating role of institutional capacity in the relationship between tax policy and income inequality. The main issues identified in the paper are the following:
-in the methodology chapter, the models used are not sufficiently presented;
-in the results chapter, the results presented in the tables are not substantiated (example: page 9, lines 316-318);
-we did not find the limitations of the study in the conclusions.
Author Response
Dear Reviewer,
We are indeed grateful to the Editor and the anonymous reviewers for their insightful comments that have enriched our manuscript.
In the main text, we have duly revised the paper following the reviewers’ comments. The revised portions are in track changes marked RED for your convenience. The details of how we have addressed the comments are as follows:
Comments |
Responses |
Reviewer #2 |
|
The research is of interest because it examines the moderating role of institutional capacity in the relationship between tax policy and income inequality. |
We thank the reviewer for this commendation. |
The main issues identified in the paper are the following: |
|
In the methodology chapter, the models used are not sufficiently presented
|
Thank you for this insightful comment.
However, to the best of our knowledge, we have provided the necessary information required to conduct an efficient and superior econometric analysis such as System Generalized Method of Moments (GMM). Notably, we sufficiently addressed the econometric model in line with recent development in the literature on income inequality (see Odusola, 2017; Salotti & Trecroci, 2018; Clifton et al., 2020; Cevik & Correa-Caro, 2020; Kunawotor et al., 2020; Brinca et al., 2021; Kunawotor et al., 2022). These active studies guided the conduct of the current paper. Nevertheless, we shall be grateful if the reviewer could point to some specific areas in the methodology that needs improvement for us to work. We are committed to improving the quality of this paper. At the moment, we find the comment to be too broad. |
In the results chapter, the results presented in the tables are not substantiated (example: page 9, lines 316-318) |
We are grateful to the reviewer for this close observation. We have addressed the comment. Please, see page 9, lines 316-318. |
We did not find the limitations of the study in the conclusions. |
We are grateful for this comment. We have mentioned the limitation of the study in the conclusion section as requested. Please, see page 15. |
For your convenience, the article related information is provided below:
Journal: Economies
Manuscript ID: economies-1667406
Title: Fiscal policy and income inequality: The critical role of
institutional capacity
Authors: Manwar Hossein Malla *, Pairote Pathranarakul
Regards,
Manwar Hossein Malla
Deputy Secretary, Finance Division
Ministry of Finance
Bangladesh Secretariat
Dhaka 1000.
Cell Phone: +8801713123278
E-Mail: manwarhm@yahoo.com
Please see the attachment (PDF File)

Round 2
Reviewer 2 Report
I appreciate the effort of the authors, I propose the publication of the article in its present form.